# Factors associated with Covid-19 vaccine acceptance among persons with disabilities: A cross-sectional study in Ghana

Godfred Atta-Osei[1]*, Enoch Acheampong[1], Daniel Gyaase[2,3], Rebecca Tawiah[1], Theresah Ivy Gyaase[1], Richard Adade[1], Douglas Fofie[4], Isaac Owusu[1], Wisdom Kwadwo Mprah[1]

1 Kwame Nkrumah University of Science and Technology, Department of Health Promotion and Disability Studies, Centre for Disability and Rehabilitation Studies, Kumasi, Ghana, 2 Injury Division, The George Institute for Global Health, Barangaroo, Australia, 3 University of New South Wale, Sydney, Australia, 4 Department of Education, Akrokerri College of Education, Akrokerri, Ghana

* attaoseig1@gmail.com

**Data Availability Statement:** All relevant data are within the paper and it Supporting Information files.

## Abstract

### Background

While COVID-19 has had a wide-ranging impact on individuals and societies, persons with disabilities are uniquely affected largely due to secondary health conditions and challenges in adhering to protective measures. However, research on COVID-19 and vaccine acceptance has primarily focused on the general population and healthcare workers but has specifically not targeted PwDs, who are more vulnerable within societies. Hence, this study assessed PwDs knowledge of COVID-19 and factors associated with COVID-19 vaccine acceptance.

### Methods

A cross-sectional survey was conducted among PwDs in the Atwima Mponua District in the Ashanti Region of Ghana. Respondents were sampled systematically and data was collected using a structured questionnaire. The data were analyzed with STATA version 16.0. Descriptive analysis was done using means and proportions. The chi-square test and Logistic regression were used to assess Covid-19 vaccine acceptance among the respondents.

### Results

250 PwDs were recruited for the study. A higher proportion of the respondents were females, physically impaired, and between 30–50 years. The majority (74%) of the PwDs had average knowledge about Covid-19. Factors such as age, educational level and type of disability were significantly associated with PwDs' knowledge of COVID-19. The acceptance rate for COVID-19 among PwDs was 71.2%. Age, religion, knowledge of COVID-19, and educational level were significantly associated with Covid-19 vaccine acceptance. Persons with disabilities with low and average knowledge of COVID-19 were 95% and 65%, respectively, less likely to accept the vaccine compared to those with high knowledge of COVID-19

**Funding:** The authors received no specific funding for this work.

**Competing interests:** The authors have declared that no competing interests exist.

(AOR = 0.05, 95%CI: 0.01, 0.21; AOR = 0.35, 95%CI: 0.12, 1.03). Older people and those with higher education were more likely to accept the vaccine compared to younger people and those with no or less education.

## Conclusion

Persons with disabilities have average knowledge of COVID-19 and a greater percentage of them were willing to accept the vaccine. The study identified age, religion, knowledge of COVID-19, and educational level as contributing factors to their willingness to accept the COVID-19 vaccine. This suggest that PwDs will lean positive toward COVID-19 vaccine programs and as such, vaccination programs should target them.

## Introduction

The coronavirus disease (Covid-19) pandemic has and continues to inflict unimaginable harm to individual, families and the economies of all nations. Leaders around the world implemented strict precautionary actions to control the virus, such as social distancing, isolation and quarantine, community containment, national lockdowns, use of face masks, regular handwashing, use of hand sanitizers and travel restrictions, to contain its spread [1, 2]. It was evident earlier in the outbreak that marginalised groups, such as persons with disabilities, the aged, people with underlining health conditions faced more challenges from covid-19 [3–5].

Research shows that persons with disabilities are at higher risk of contracting the Covid-19 than those without disabilities precisely because of secondary health conditions [6]. Persons with disabilities who are institutionalized for care and rehabilitation for example are particularly vulnerable to Covid-19 since some of these facilities have limited space to allow for social distancing [7]. Comas-Herrera et al., in their preliminary studies, observed that, of all Covid-19 deaths, 42% to 57% were reported in care homes [8]. Additionally, many persons with disabilities find it difficult to comply with protective measures, such as physical distancing, effective handwashing, or mask-wearing [9]; some people with disabilities require their caregivers to perform some activities of daily living for them and as a result, practicing social distancing is difficult [6, 10, 11].

According to Pogue et al., vaccines are the most successful way of dealing with infectious diseases such as the Covid-19 pandemic and their spread [12]. Scientists worldwide made efforts toward developing safe and efficacious vaccines to curb the spread of the disease [12]. Despite the enormous efforts made toward Covid-19 vaccine development and use, acceptance and hesitancy remain a serious roadblock. Researchers have studied the acceptance of the Covid-19 vaccine and have reported varying findings from different countries and populations. Covid-19 vaccine acceptance rate in Africa ranges from 43.55% in Egypt to 82.76% in Mauritius [13].

In Ghana, Acheampong et al. in 2021 found a 51% acceptance rate of Covid-19 vaccine [14]. Among Ghanaian health workers' acceptability of Covid-19 vaccines, about 60% were unwilling to receive the Covid-19 vaccine [15].

Studies on knowledge of Covid-19 and vaccine acceptance have mostly focused on the general public and healthcare workers. To now, and to the best of our knowledge, no study on Covid-19 vaccine acceptance in Ghana has targeted persons with disabilities, who are also disproportionately at risk of Covid-19 infections. Hence, the current study assessed the knowledge of Covid-19 and factors associated with its vaccine acceptance among persons with disabilities.

## Methods

### Study design

The study employed an analytical cross-sectional design to assess Covid-19 vaccine acceptance among persons with disabilities.

### Setting

The study was conducted in the Atwima Mponua district. The district is located in the south-western part of the Ashanti region and is the second largest district (1,883.2 square kilometers) in the region. As of 2010, the number of persons with disabilities in this district was 2686 [16]. The district was not recognized as a Covid-19 hotspot, and as a result, the vaccination programme that was being rolled out was made available only to essential workers and not made available to the population at the time of data collection.

### Population and sample size

The population of the study was persons with disabilities who had registered with the department of social welfare at the Atwima Mponua District Assembly. On record, 671 persons with disabilities comprising 413 physically impaired, 148 visually impaired, 63 hearing impaired, and 49 other impairment groups (intellectual, autism, albinism, among others) had registered with the department of social welfare at the Atwima Mponua district assembly. The Yamane formula for calculating sample size was used to estimate the sample size.

$$n = \frac{N}{1 + Ne^2}$$

Where n = sample size, N = Population and e = margin of error

With a population of 671 and a margin of error of 5%, the sample size was estimated to be 250.

### Sampling technique

Systematic sampling technique was used to sample respondents for the study. The population was divided into strata based on the type of disability. Respondents were sampled randomly from each of the strata based on sampling proportion to size. The number of each impairment type and the number sampled are shown in Table 1 below.

### Study variables

The outcome variable for the study was Covid-19 vaccine acceptance which was measured as a dichotomous variable, thus "Yes = 1" or No = 0". The independent variables were persons with disabilities' knowledge of Covid-19 and their demographic characteristics such as age, sex, type of impairment, marital status, employment status, level of income and religious affiliation. Knowledge

**Table 1. Distribution of impairment types and number sampled.**

| Impairment type | Number on register | Proportion to size |
| --- | --- | --- |
| Physical | 413 | 154 |
| Visual | 148 | 55 |
| Hearing | 63 | 23 |
| Other | 49 | 18 |
| **Total** | **671** | **250** |

of Covid-19 was assessed with 11 statements using "Yes = 1" or No = 0" responses. Each correct answer to the 11 questions was scored 1 and 0 for the wrong answer. The questions were summed up to generate a knowledge score with a maximum score of 11 and a minimum of 0. The score was further categorized into High, Average and Low as the level of knowledge on Covid-19.

### Data collection

A structured questionnaire made up of three sections was administered to respondents. Section one comprised respondents' socio-demographic characteristics such as age, gender, impairment type, level of education, marital status, employment status, level of income, and religious background. Section two featured the knowledge of Covid-19. Section three which was only one statement, measured whether or not PwDs will accept Covid-19 Vaccine. The data were collected between August and September 2021. Three members of the research team comprising two with expertise in Ghanaian Sign Language Interpretation administered the questionnaires to respondents in their communities. Some respondents opted to complete the questionnaires themselves with guidance from the researchers. For those who could not complete the questionnaire on their own, the researchers read and explained each statement on the questionnaire in Asante Twi language to their understanding before making a choice. The researchers signed statements on the questionnaire in Ghanaian sign language to the understanding of deaf participants. They were allowed some time to make their choice before moving to the next statement.

### Statistical analysis

Data analysis was carried out using STATA version 16. Descriptive statistics were used to summarize the variables in the study. Means and standard deviation were estimated for continuous variables, while frequencies and percentages were performed on categorical variables. The Chi-Square test was used to examine socio-demographic factors influencing PwDs knowledge of covid-19. Variables with a p-value of 0.05 or less were considered significant. Univariate and multivariable logistic regression was used to identify factors influencing covid-19 vaccine acceptance among PwDs. Independently significant variables ($p \leq 0.2$) at the univariate level were added to the multivariable model using the forward stepwise approach. Also, variables of significant importance from other studies on vaccine acceptance that were not significant in the univariate analysis were accounted for in the multivariate model. The likelihood ratio test and the AIC were used in assessing the goodness of fit of the models. Confidence intervals were also computed at a 95% confidence level, and variables with a p-value $\leq 0.05$ were considered statistically significant in the final model.

### Ethical approval

Ethical approval was obtained from the Kwame Nkrumah University of Science and Technology's (KNUST) Committee on Human Research and Publication Ethics (CHRPE) with Reference number CHRPE/AP/359/21. A letter of approval was also obtained from the Atwima Mponua District Assembly. The study aims and objectives were explained to respondents, and consent forms were given to either sign or thumbprint. They were also assured strict anonymity and confidentiality of the data they would provide.

## Results

### Demographic characteristics of respondents

As shown in Table 2 below, a total of 250 persons with disabilities were recruited for this study. The minimum age of respondents was 18 and the maximum age was 90. Their

**Table 2. Demographic characteristics of the respondents.**

| Variable | Frequency | Percentage (%) |
|---|---|---|
| **Age** | | |
| 18–35 | 65 | 26.0 |
| 36–50 | 71 | 28.4 |
| 51–65 | 53 | 21.2 |
| 66–80 | 46 | 18.4 |
| 80+ | 15 | 6.0 |
| **Gender** | | |
| Male | 115 | 46.0 |
| Female | 135 | 54.0 |
| **Impairment Type** | | |
| Physical | 154 | 61.6 |
| Visual | 55 | 22.0 |
| Hearing | 23 | 9.2 |
| Other | 18 | 7.2 |
| **Level of Education** | | |
| None | 92 | 36.8 |
| Basic | 113 | 45.2 |
| Secondary | 32 | 12.8 |
| Tertiary | 13 | 5.2 |
| **Marital Status** | | |
| Married | 122 | 48.8 |
| Single | 56 | 22.4 |
| Widowed | 39 | 15.6 |
| Divorced | 31 | 12.4 |
| Cohabiting | 2 | 0.8 |
| **Employment Status** | | |
| Employed | 100 | 40.0 |
| Unemployed | 143 | 57.2 |
| Student | 7 | 2.8 |
| **level of income** | | |
| Less than Ghc.500 | 216 | 86.4 |
| 500 to Ghc.1000 | 24 | 9.6 |
| More than Ghc.1000 | 10 | 4.0 |
| **Religious Affiliation** | | |
| Christian | 217 | 86.8 |
| Islam | 31 | 12.4 |
| Other | 2 | 0.8 |

average age was found to be 50.2 with a standard deviation of 19.5. A higher proportion, 61.6% of the respondents, were physically impaired, 22.0% visually impaired, 9.2% hearing impaired, and other impairment groups 7.2%. A little above half, 54% were female, 28.4% were aged 36–50 years, and about 45% had attained basic education. On marital status, 48.8% were married, while 22.4% were single. A higher proportion of the respondents, 57.2%, were unemployed, with about 3% being students. A majority (86.4%) of the respondents had a monthly income of less than 500 Ghana Cedis (USD 83, as of 2021), and 86.4% were Christians.

## Knowledge on Covid-19 among respondents

Table 3 presents the knowledge of Respondents on Covid-19 outbreak. Generally, study observed that respondents have good knowledge of Covid-19. About 14.8 respondents had high knowledge of the Covid-19 and majority (74.0%) of the respondents had average knowledge of Covid-19. For example, most (88%) respondents stated that Covid-19 is a viral infection, while a few stated otherwise. About 86.4% of the respondents indicated that Covid-19 is transmitted by close contact with an infected person. Regarding Covid-19 being declared as epidemic, 170 respondents representing 68%, responded no, meanwhile Covid-19 is declared as a pandemic.

A percentage majority (75.6%) of the respondents indicated that the incubation period of Covid-19 is one month, which is not true and a little above half of the respondent (54.8%) indicated that Covid-19 is curable. On the transmission of the virus, 89.6 stated that visiting crowded places and being close to people showing symptoms increase ones' risk of Covid-19 infection and handwashing with soap was indicated by the majority (90.8) as a safety protocol to prevent the spread of the disease.

## Factors influencing PwDs' level of knowledge of Covid-19

Table 4 shows some influencing factors on PwDs' level of knowledge on Covid-19. Analysis revealed that an association exists between factors such as age, type of disability, level of education and PwDs level of knowledge on Covid-19. Respondent's age was significantly associated with their level of knowledge of Covid-19 (p = 0.014). A higher proportion of respondents in all age groups had an average knowledge of Covid-19 (ranging from 53% among those aged 80 + years to 83% among those aged 51–65 years). The type of disability was found not to have statistical significance on respondent's level of knowledge on Covid-19 (p = 0.055), with people with visual impairment having a greater percentage (20%) of higher knowledge of Covid-19 while those with hearing impairment had a higher percentage (26%) of low knowledge of Covid-19. There was an increasing order of high levels of knowledge with increasing

**Table 3. Knowledge of respondents on COVID-19.**

| Questions | Frequency | Percentage | Frequency | Percentage |
|---|---|---|---|---|
| | *Yes* | *%* | *No* | *%* |
| Covid-19 is a viral infection | 220 | 88.0 | 30 | 12.0 |
| Covid-19 is transmitted by close contact with an infected person | 216 | 86.4 | 34 | 13.6 |
| Covid-19 is declared an epidemic by WHO | 80 | 32 | 170 | 68.0 |
| Covid-19 does not affect children | 91 | 36.4 | 159 | 63.6 |
| Fever, dry cough, shortness of breath, and vomiting are symptoms of Covid-19 | 151 | 60.4 | 99 | 39.6 |
| The incubation period for Covid-19 is one month | 189 | 75.6 | 61 | 24.4 |
| Covid-19 is not curable | 113 | 45.2 | 137 | 54.8 |
| The elderly and individuals with underlying chronic diseases have a greater risk of developing severe symptoms from Covid-19 infection | 216 | 86.4 | 34 | 13.6 |
| Visiting crowded places and being close to people showing symptoms increase the risk of Covid-19 | 224 | 89.6 | 26 | 10.4 |
| Washing hands with water and soap can help in the prevention of Covid-19 transmission | 227 | 90.8 | 23 | 9.2 |
| Covid-19 does not affect pregnant women. | 111 | 44.4 | 139 | 55.6 |
| **Knowledge of Covid-19** | | | | |
| Low | 28 | 11.2 | | |
| Average | 185 | 74.0 | | |
| High | 37 | 14.8 | | |

**Table 4. Demographic factors influencing PwDs level of knowledge on Covid-19.**

| Variables | High (%) | Average (%) | Low (%) | P-value |
|---|---|---|---|---|
| **Age** | | | | **0.014** |
| 20–35 years | 9 (13.) | 45 (69.2) | 11 (16.9) | |
| 36–50 years | 13 (18.3) | 54 (76.1) | 4 (5.6) | |
| 51–65 years | 3 (5.7) | 44 (83.0) | 6 (11.3) | |
| 66–80 years | 10 (21.7) | 34 (73.9) | 2 (4.4) | |
| 80+ | 2 (13.8) | 8 (53.3) | 5 (11.2) | |
| **Sex** | | | | 0.769 |
| Male | 19 (16.5) | 83 (72.2) | 13 (11.3) | |
| Female | 18 (13.3) | 102 (74.0) | 15 (11.2) | |
| **Type of disability** | | | | 0.055 |
| Physical | 23 (14.9) | 120 (77.9) | 11 (7.2) | |
| Visual | 11 (20.0) | 35(63.22) | 9 (16.4) | |
| Hearing | 2 (8.7) | 15(65.2) | 6 (26.1) | |
| Other | 1 (5.6) | 15(83.3) | 2(11.1) | |
| **Level of Education** | | | | **0.039** |
| None | 10 (10.9) | 72 (76.3) | 10 (10.9) | |
| Basic | 14 (12.4) | 85 (75.2) | 14 (12.4) | |
| Secondary | 7 (21.9) | 22 (68.8) | 3 (9.4) | |
| Tertiary | 6 (46.2) | 6 (46.2) | 1 (7.7) | |
| **Employment status** | | | | 0.189 |
| Employed | 15 (15.0) | 78 (78.0) | 7 (7.0) | |
| Unemployed | 20 (14.0) | 104 (72.8) | 19 (13.3) | |
| Student | 2 (28.6) | 3 (42.9) | 2 (28.6) | |
| **Level of income** | | | | 0.089 |
| Less than 500 | 29 (13.4) | 162 (75.0) | 25 (11.6) | |
| 500 to 1000 | 4 (16.7) | 19 (79.2) | 1 (1.2) | |
| More than 1000 | 4 (40.0) | 4(40.0) | 2 (20.0) | |
| **Religious affiliation** | | | | 0.271 |
| Christian | 34 (15.7) | 157 (72.4) | 26 (12.0) | |
| Islam | 2 (6.5) | 27 (87.1) | 2 (6.5) | |
| Other | 1 (50.0) | 1 (5.0) | 0 (0.00) | |

educational level (11% for those without education to 46% among those with tertiary education) with significant association (p = 0.039).

## Factors influencing acceptability of Covid-19 vaccine among PwDs

Table 5 presents the crude and adjusted logistic regression of factors influencing PwDs willingness to accept Covid-19 vaccine. From the crude analysis, PwDs aged 36–50 years and 51–65 years are 3.1 times and 3.7 times respectively more likely to accept Covid-19 vaccine compared to those between 20–35 years (COR = 3.08, 95%CI = 1.44–6.61; COR = 3.70, 95%CI = 1.55–8.82 respectively). After adjusting for other variables, those aged between 36–50 years and 51–65 years had 3.3 and 5.2 odds of accepting the Covid-19 vaccine compared to those within the age group 20–35 years, respectively holding all other variables constant (AOR = 3.28, 95%CI; 1.37–7.84; AOR = 5.24, 95%CI; 1.91, 14.36).

With regards to their knowledge level on Covid-19 and vaccine acceptability, those with low knowledge were 91% less likely to accept the vaccine compared to those with high

**Table 5. Logistic regression of factors influencing PwDs willingness to accept Covid-19 vaccine.**

| Variables | Crude | | | Adjusted | | |
|---|---|---|---|---|---|---|
| | COR | 95%CI | P-value | AOR | 95%CI | P-value |
| **Age** | | | | | | |
| 20–35 years | | | | | | |
| 36–50 years | 3.08 | 1.44, 6.61 | **0.004** | 3.28 | 1.37, 7.84 | **0.008** |
| 51–65 years | 3.70 | 1.55, 8.82 | **0.003** | 5.24 | 1.91, 14.36 | **0.001** |
| 66–80 years | 1.42 | 0.65, 3.10 | 0.380 | 0.97 | 0.40, 2.32 | 0.941 |
| 80+ | 1.51 | 0.46, 4.93 | 0.491 | 2.05 | 0. 50, 8.40 | 0.320 |
| **Religion** | | | | | | |
| Christianity | | | | | | |
| Islam | 0.38 | 0.18, 0.82 | **0.013** | 0.28 | 0.12, 0.67 | **0.004** |
| **Knowledge on Covid-19** | | | | | | |
| High | | | | | | |
| Average | 0.43 | 0.16, 1.18 | 0.101 | 0.35 | 0.12, 0.89 | **0.047** |
| Low | 0.09 | 0.03, 0.29 | **<0.001** | 0.05 | 0.01, 0.21 | **<0.001** |
| **Employment status** | | | | | | |
| Employed | | | | | | |
| Unemployed | 1.45 | 0.82, 2.56 | 0.198 | 2.06 | 1.03, 4.12 | **0.040** |
| Student | 0.19 | 0.03, 1.02 | **0.053** | 0.32 | 0.05, 2.16 | 0.241 |
| **Type of disability** | | | | | | |
| Physical | | | | | | |
| Visual | 0.57 | 0.30, 1.12 | 0.099* | | | |
| Hearing | 0.75 | 0.29, 1.96 | 0.555 | | | |
| Other | 0.51 | 0.19, 1.42 | 0.200 | | | |

COR = Crude Odds Ratio, AOR = Adjusted Odds Ratio, CI = Confidence Interval

**Note**: The adjusted model included all variables in the table.

Muslims were 62% less likely to accept the Covid-19 vaccine compared to their counterparts, the Christians in the crude analysis (COR = 0.38, 95CI%: 0.18, 0.82). After controlling for other variables, Muslims were 72% less likely to accept the Covid-19 vaccine compared to Christians (AOR = 0.28, 95CI%: 0.12, 0.67).

knowledge (COR = 0.09, 95CI%: 0.03, 0.29), whiles those with average knowledge were 57% less likely to accept the vaccine compared to those with higher knowledge but this was not statistically significant (COR = 0.43, 95%CI: 0.16, 1.18). After adjusting for other variables and holding all variables constant, those with low knowledge of Covid-19 were 95% less likely to accept the vaccine (AOR = 0.05, 95%CI: 0.01, 0.21) while those with average knowledge were 65% less likely to accept the vaccine compared to those with high knowledge on Covid-19 (AOR = 0.35, 95%CI: 0.12, 1.03).

For employment status, the univariate analysis shows that students were 81% less likely to accept the vaccine compared to those who are employed (COR = 0.19, 95%CI: 0.03, 1.02). After controlling for other variables, students were 68% less likely to accept the vaccine compared to those who were employed, but this was not statistically significant. PwDs who were not employed were 2.1 times more likely to accept the Covid-19 vaccine compared to those who were employed (AOR = 2.06, 95%CI: 1.03, 4.12).

## Discussion

Our current study found a higher knowledge of covid-19 among the respondents. Though studies in Ghana have assessed the general public knowledge of Covid-19 [17–19], this is the first study to report on the persons with disabilities' knowledge of the pandemic. The finding

of this study is similar to findings from studies with the general public [17–19], which reported a higher level of knowledge on Covid-19. Our study revealed an average 70.00% knowledge score among respondents. The finding corroborates a study conducted among the general public in China by Pakpour and colleagues, where a high knowledge of Covid-19 was observed among respondents [20]. The high knowledge of Covid-19 among persons with disabilities might be attributed to the inclusiveness of education on the subject including the regular addresses by the president of Ghana.

Factors such as age, type of disability, level of education and income were found to influence respondents' knowledge of Covid-19. This finding is similar to a study in Ghana by Kwabla et al. which also identified age and education as positive influencing factors on people's knowledge of covid-19 [17]. The type of impairment was also found to influence the respondent's level of knowledge. The visually and physically impaired were found to have higher knowledge than those with hearing loss. This disparity in knowledge is probably because not all the educational programmes had sign language interpretation services, and not all the deaf people had television set to 'listen' to the president's addresses, which did not contain all the information on Covid-19 management. Also, the majority of people in rural settings, such as where this study was conducted in Ghana, mostly rely on radio stations for information. These radio stations report in audio form, which is also not accessible for persons with hearing loss. This could be the reason why those with hearing impairment had less knowledge.

Studies have found varying rates of Covid-19 vaccine acceptance across countries; 29.4% in France [21], 85.36% in Brazil, 81.58% in South Africa, 80% in Denmark, 79% in the UK [22], 55.9% in Democratic Republic of Congo [23]. In Ghana, studies have reported varying acceptance rates among different populations, 58.2% in the general public [24] and 39.3% among healthcare workers [15]. In this study among respondents, were found to have a higher (71.2%) acceptance to Covid-19 vaccines. The differences observed in the current study and other studies could be attributed to the difference in the studies' times. Earlier in the pandemic, people's willingness to accept vaccines under development might be lower than when they were approved for use. The study was, however, not consistent with Shekhar et al. study among minority groups (black HealthCare professionals) which estimated a 19% covid-19 vaccine acceptance [25]. The findings from the current study portray a good indication for the implementation of a successful vaccination program among PwDs.

The study identified age, religion, employment status and level of knowledge as factors significantly associated with Covid-19 vaccine acceptance. Machida et al. reported that the older the person, the more likely he/she is likely to receive a Covid-19 vaccine which is consistent to findings in the current study [26]. Older people were one of the most at risk groups for Covid-19, with a higher fatality rate. Most of the underlining health conditions for covid-19 susceptibility, such as hypertension, diabetes, etc., are also age-related, which could be the reason why older people were most likely to accept the vaccine.

The study also found an association between Covid-19 vaccine acceptance and religion. Christians were more likely to accept the vaccine than Muslims. Though this finding supports the assertion by Khan et al. that false claims are spread among Muslim communities on vaccine development [27], it might not be the case in this study. Muslims were about 12% compared to over 87% Christians, making the data skewed to the Christian religion.

Knowledge of health conditions or pandemics are seen as one of the factors associated with vaccine acceptance. For example, Setbon and Raude observed that inadequate knowledge is one of the reasons for vaccine hesitancy [28]. Our study found that the level of knowledge one has determines their acceptance of the vaccine. Similar to what was observed in this study, a study by Mohamed et al. in Malaysia also reported higher acceptance of Covid-19 vaccine among people with higher knowledge of the pandemic [29]. However, a multi-regional study

among the general public in Ghana reported a contradictory result [30]. Their study did not find any significant association between vaccine acceptance and knowledge of Covid-19. During the early stages, the Covid-19 and its vaccines were associated with a lot of misinformation in Ghana. For example, it was believed that the vaccine would make people impotent or barren. This belief could influence people's acceptability of the vaccine, but it appears that most of the respondents in this current study were not influenced by the misinformation.

### Limitation

The study is limited to persons with disabilities in the Atwima Mponua District Assembly who have registered with the department of social welfare. This means that the views of PwDs in the district who have not registered with department of social welfare were not captured in the current study. Also, the study was dominated by the respondents from the Christian religion (87%) which made the findings skewed. Again, the study was conducted in a rural area which might not be the case in the urban setting. We therefore recommend that future study should include a sizeable proportion of respondents from other religions; target those who are not registered with the department of social welfare and compare the results with this study.

### Recommendations

Since PwDs constitute the most vulnerable population in society in terms of the infection and negative consequences of the Covid-19 vaccines and also willing to accept Covid-19 vaccines, efforts must be made by the ministry of health and the Ghana Health Service to prioritise them for Covid-19 vaccination. Also, there is a high knowledge on Covid-19 and its vaccines among persons with disabilities, hence, we recommend that the Ghana health service should sustain health promotion campaigns and behaviour change communication on Covid-19 to enable people to accept Covid-19 vaccines. It is also recommended that the media such as the radio stations and TV stations should continue their education and discussion of Covid-19 and the importance of vaccination.

### Conclusion

Our study showed that knowledge of covid-19 was high among respondents, and they were willing to accept Covid-19 vaccines. Demographic characteristics such as age, type of disability, and level of education were associated with Covid-19 vaccine acceptance. Also, the study found that an increase in the knowledge of Covid-19 among PwDs increases their acceptance of its vaccine.

### Supporting information

**S1 Data.**
(DO)

**S2 Data.**
(DTA)

### Acknowledgments

We are grateful to the Leadership of the Atwima Mponua District Assembly for giving us the approval to work in the district. we are also grateful to the director of Social Welfare and Community Development, Mr. Carlton D. Mawulorm for his support in getting approval from the district assembly and Mr. Augustine Akwaboah (National Service Personnel at the department

of social welfare as at time of data collection) for his support in the data collection process. We are also particularly grateful to all persons with disabilities who availed themselves for this study.

## Author Contributions

**Conceptualization:** Godfred Atta-Osei, Enoch Acheampong, Daniel Gyaase.

**Data curation:** Godfred Atta-Osei, Rebecca Tawiah, Theresah Ivy Gyaase, Douglas Fofie.

**Formal analysis:** Godfred Atta-Osei, Daniel Gyaase, Rebecca Tawiah, Richard Adade.

**Methodology:** Enoch Acheampong, Daniel Gyaase, Theresah Ivy Gyaase, Richard Adade, Douglas Fofie, Isaac Owusu, Wisdom Kwadwo Mprah.

**Supervision:** Enoch Acheampong, Isaac Owusu, Wisdom Kwadwo Mprah.

**Validation:** Enoch Acheampong, Daniel Gyaase, Isaac Owusu, Wisdom Kwadwo Mprah.

**Writing – original draft:** Godfred Atta-Osei, Daniel Gyaase, Rebecca Tawiah, Theresah Ivy Gyaase, Richard Adade.

**Writing – review & editing:** Godfred Atta-Osei, Enoch Acheampong, Daniel Gyaase, Rebecca Tawiah, Theresah Ivy Gyaase, Richard Adade, Douglas Fofie, Isaac Owusu, Wisdom Kwadwo Mprah.

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
