## [Decision Letter · Decision Letter 0]

17 Oct 2023

PGPH-D-23-01529

Factors Associated with Covid-19 Vaccine Acceptance among Persons with Disabilities: A cross-sectional study in Ghana

Dear Dr. Atta-Osei,

Thank you for submitting your manuscript to PLOS Global Public Health. After careful consideration, we feel that it has merit but does not fully meet PLOS Global Public Health’s publication criteria as it currently stands. Therefore, we invite you to submit a revised version of the manuscript that addresses the points raised during the review process.

We look forward to receiving your revised manuscript.

Kind regards,

Abram L. Wagner, PhD, MPH

Academic Editor

Journal Requirements:

Additional Editor Comments (if provided):

Reviewer 1 has some minimal comments to address which I believe would improve paper.

Reviewers' comments:

Reviewer's Responses to Questions

**Comments to the Author**

1. Does this manuscript meet PLOS Global Public Health’s publication criteria? Is the manuscript technically sound, and do the data support the conclusions? The manuscript must describe methodologically and ethically rigorous research with conclusions that are appropriately drawn based on the data presented.

Reviewer #1: Yes

Reviewer #2: Yes

2. Has the statistical analysis been performed appropriately and rigorously?

Reviewer #1: I don't know

Reviewer #2: Yes

3. Have the authors made all data underlying the findings in their manuscript fully available (please refer to the Data Availability Statement at the start of the manuscript PDF file)?

Reviewer #1: Yes

Reviewer #2: Yes

4. Is the manuscript presented in an intelligible fashion and written in standard English?

Reviewer #1: Yes

Reviewer #2: Yes

5. Review Comments to the Author

Reviewer #1: Line 15-16: (research on Covid-19 vaccine acceptance has not targeted PwDs) -Before you write this, you should state whether there has been research among other groups. Also the justification of the study should be clearly stated…That is, the rationale of this study

Line 18 - 21: Try to rewrite this section. It is not comprehensive enough, and the method should include; study design, study population, sampling techniques, data collection (you can include the questionnaire/research tool used), and data analysis

Line 22 - 31: Start the result section with the descriptive analysis (like mean age, age range, etc) before proceeding to the inferential analysis result

Line 31 - 35 The sentence is too long, and looks confusing…try to rephrase it.

Line 46: Be consistent. Are you using the term ‘coronavirus” or ‘Covid-19’...Stick with one

Line 109 - 110 You should not mention that the questionnaire was made up of six sections since this study only focused on two sections. You can just state that the questionnaire used in this study was made up of two sections

Line 112 - 113 But why is section two measuring both the independent and dependent variables…Each section should stand for each variable

Line 131 - 133: Start with the mean age, then age range, and others

Line 141: The percentage should be in one decimal place

Line 141 -158: Try to rephrase the write-up. Also, be consistent with the flow of table. Do not back and forth on the table, report the result chronologically, as it appears in the table

Line 147: (15%) - State the exact value. 14.8% not 15%; condition - What condition? Try to state it; the average correct response was 70% - Where did this come from

Line 153: 90% - Use the exact value…do not round up

Line 160 - 172: For this section, you do not need to write about the percentage. You should only be concerned with the p-value and association

Line 161: Analysis - State the analysis used

Line 166 - 167: The type of disability was also found to have statistical significance on respondent’s level of knowledge on Covid-19 (p=0.055) - It was not significant. 0.055 is not significant

Line 177 - 181: You do not need to write everything from the table; but if you want to write it, you can try to put the sentences together as one e.g PwDs aged 36-50 years and 51-65 years are 3.1 times and 3.7 times respectively more likely to accept Covid-19 vaccine compared to those between 20-35 years (COR=3.08, 95%CI=1.44-6.61; COR=3.70, 95%CI=1.55-8.82 respectively).

Line 189 - 203: Follow the format of the comment made above

Line 209 - 217: This is not necessary…It should be removed

Line 277 - 278: There is a section of persons with disabilities who have not registered with the department - Rephrase this

Reviewer #2: The authors may consider revising sentence on page 20 line 213 as follows: "Measures taken against the spread of Covid-19 virus"

Also a clear description of method employed in interviewing persons with hearing impairment could clarify the results as presented

6. PLOS authors have the option to publish the peer review history of their article (what does this mean?). If published, this will include your full peer review and any attached files.

**Do you want your identity to be public for this peer review?** For information about this choice, including consent withdrawal, please see our Privacy Policy.

Reviewer #1: **Yes: **Isaac Iyinoluwa Olufadewa

Reviewer #2: **Yes: **OSCAR BANGRE

---

## [Editor Report · Decision Letter 1]

30 Jan 2024

PGPH-D-23-01529R1

Factors Associated with Covid-19 Vaccine Acceptance among Persons with Disabilities: A Cross-sectional Study in Ghana

Dear Dr. Atta-Osei,

Thank you for submitting your manuscript to PLOS Global Public Health. After careful consideration, we feel that it has merit but does not fully meet PLOS Global Public Health’s publication criteria as it currently stands. Therefore, we invite you to submit a revised version of the manuscript that addresses the points raised during the review process.

We look forward to receiving your revised manuscript.

Kind regards,

Abram L. Wagner, PhD, MPH

Academic Editor

Journal Requirements:

Additional Editor Comments (if provided):

I have minor formatting issues I would like you to process before issuing an acceptance.

Can you be consistent in your use of PWD vs PwD (I think just abstract has PWD)?

In Table 4, could you switch order of confidence intervals and P-values?

In Table 4 can you make a note of what variables are included in the adjusted model? (Even if it's a footnote saying something like: "Adjusted analysis includes all covariates listed in this table.")

I will note that stepwise selection procedures are not very recommended anymore. If you want to keep it in, that is fine, but a better approach may be to a priori decide on a list of covariates to include in the model, and to just adjust for all of those, regardless of bivariate significance.
---

## [Editor Report · Decision Letter 2]

15 Feb 2024

Factors Associated with Covid-19 Vaccine Acceptance among Persons with Disabilities: A Cross-sectional Study in Ghana

PGPH-D-23-01529R2

Dear Mr Atta-Osei,

We are pleased to inform you that your manuscript 'Factors Associated with Covid-19 Vaccine Acceptance among Persons with Disabilities: A Cross-sectional Study in Ghana' has been provisionally accepted for publication in PLOS Global Public Health.

Best regards,

Abram L. Wagner, PhD, MPH

Academic Editor